# Phytochemical Analysis of *Anastatica hierochuntica* and *Aerva javanica* Grown in Qatar: Their Biological Activities and Identification of Some Active Ingredients

**DOI:** 10.3390/molecules28083364

**Published:** 2023-04-11

**Authors:** Vandana Thotathil, Naheed Sidiq, Ameena Fakhroo, Lakshmaiah Sreerama

**Affiliations:** Department of Chemistry and Earth Sciences, Chemistry Program, College of Arts and Sciences, Qatar University, Doha P.O. Box 2713, Qatarafakhro2@qu.edu.qa (A.F.)

**Keywords:** *Anastatica hierochuntica*, *Aerva javanica*, phytochemical analysis, antibacterial activities, antifungal activities, antioxidant activities

## Abstract

Plant-derived compounds and their extracts are known to exhibit chemo preventive (antimicrobial, antioxidant and other) activities. The levels of such chemo preventive compounds vary depending on environmental factors, including the regions where they grow. Described in this study are: (i) a phytochemical analysis of the two plants grown in the desert environment of Qatar, viz., *Anastatica hierochuntica* and *Aerva javanica*; (ii) the antibacterial, antifungal and antioxidant activities of various solvent extracts of these plants; (iii) a report on the isolation of several pure compounds from these plants. The phytochemical screening indicated the presence of glycosides, tannins, flavonoids, terpenoids, saponins, phenol and anthraquinones in various extracts of each of the plants. Antibacterial and antioxidant activities were studied using agar diffusion and DPPH methods, respectively. The extracts of *Anastatica hierochuntica* as well as *Aerva javanica* inhibit the growth of both gram-positive and gram-negative bacterial species. Various extracts of the two plants also exhibited higher or similar antioxidant activities as those of the standard antioxidants, α-tocopherol and ascorbic acid. The extracts of these plants were further purified by HPLC and characterized by IR and NMR techniques. This process has led to identification of β-sitosterol, campesterol and methyl-9-(4-(3,4-dihydroxy-1′-methyl-5′-oxocyclohexyl)-2-hydroxycyclohexyl)nonanoate from *Anastatica hierochuntica*, and lupenone, betulinic acid, lupeol acetate and persinoside A and B from *Aerva javanica*. The results reported herein suggests that *Anastatica hierochuntica* and *Aerva javanica* are potent sources of phytomedicines.

## 1. Introduction

From ancient times, natural products have been considered a vital source for therapeutic agents. With evolving scientific advancement, the unique properties of these natural products have been studied and analyzed for chemical and structural diversity in relation to their therapeutic and other biological activities. Further, the medicinal plant species growing in different environments are known to have variations in their chemical constituents. Accordingly, it is important to analyze these medicinal plants as well as it is essential to preserve and maintain wild indigenous plants that are of economic and medicinal use.

As in many countries, the demand for herbal and traditional medicines are thriving in Qatar. The natural sources for these medical practices come from the limited natural vegetation and agricultural forms. *Anastatica hierochuntica* and *Aerva javanica* are among the very few plants that grow in Qatar and are part of the traditional medicine. While there are some preliminary studies on these Qatari medicinal plants, the data on isolation, characterization and their biological activities are limited [1,2].

*Anastatica hierochuntica*, also known as blooming virgin hand (locally called *Kaf Mariam* or *Jefaiea*) is a herbal medicinal plant belonging to the Brassicaceae family. Apart from Qatar, it is found in arid areas of other Arab countries such as Egypt, Saudi Arabia, Oman, United Arab Emirates, Kuwait and Iraq, as well as in some South Asian, African and European countries. The preparations of *Anastatica hierochuntica* is used to overcome fatigue, menstrual cramps, prevent uterine hemorrhage in pregnant women and facilitate smooth delivery [1,3,4]. The commonly found phytochemical constituents with biological activities identified in various parts (leaves, stem and seeds) of this plant—using ethanol, methanol, ethyl acetate, hexane and aqueous extracts—include tannins, sterols, terpenes, flavonoids, alkaloids, saponins, resins, phenols and glycosides [5]. Yoshikawa and associates have identified and isolated Anastatin A and B (flavonoids with benzofuran moiety) from the methanolic extract of *Anastatica hierochuntica* grown in Egypt [6,7]. Anastatin A and B have shown hepatoprotective activity against mouse hepatocytes and this hepatoprotective activity was significantly higher than the related flavonoids and the commercially available standard silybin [6]. Yoshikawa and associates have also reported the isolation of three new neolignans named hierochins A, B and C. These neolignans inhibit the production of nitric oxide and thus act as vasodilators under physiologic conditions [7].

Herbal tea preparations from the seeds of *Anastatica hierochuntica* (commonly consumed in Saudi Arabia) are known to exhibit antioxidant activity [8]. The compounds predominantly present in the tea preparations include phenolic acids, chlorogenic acids and flavonoids. The antioxidant activity of the herbal tea was mainly attributed to the presence of flavonoids [8,9]. On the other hand, it is also important to note that the phenolic compounds present in plants are well known to be antioxidants and free radical scavengers [8,9]. A full list of flavonoids present in *Anastatica hierochuntica* have been reported and the dominant flavonoids include flavonol aglycone, flavonol-3-*O*-glycoside, glycoflavone and two dihydroflavones [10]. The polar and non-polar phenolic compounds present in *Anastatica hierochuntica* have been recently attributed to the nephroprotective, antioxidant and free radical scavenging activities [11]. The ethanol and methanol extracts of *Anastatica hierochuntica* (grown in arid conditions of Egypt) have been shown to exhibit antibacterial activity against a number of gram-positive and gram-negative bacterial strains (*E. coli*, *S. aureus*, *P. aeruginosa*, *P. fluorescens*, *S. typhinurium* and *L. monocytogens*) [9,12,13]. The aqueous extracts, on the other hand were only effective against gram-negative bacteria such as *E. coli* and *P. fluorescens* [9,12,13]. Furthermore, none of the extracts exhibited antifungal activity when tested against *Aspergillus niger* [9]. The presence of a variety of bioactive compounds including terpenes, sterols, alkaloids, esters, thiols, fatty acids alcohols, fatty acid amines, phenols, hydrocarbons and carbohydrates have been reported from leaf, stem and seed extracts of *Anastatica hierochuntica* in hexane, ethyl acetate and methanol fractions, respectively [14].

*Aerva javanica,* known as dessert cotton (vernacular name: *Towayim*, *Tarfa*, *Tirf* or *Tuwaim*), is a medicinal plant and belongs to Amaranthaceae family. It is a perennial and an indigenous drought-tolerant herb widespread in central and southern parts of Qatar. It also grows in various parts of Asia including the Arabian Peninsula and many other parts of the world. More than 20 species of the genus *Aerva* have been reported in South Asian countries such as India and Pakistan, and most of them are used in herbal medicinal preparations [15]. Various biochemical, phytochemical and antioxidant compounds such as steroids, triterpenes, flavonoids, tannins, saponins, alkaloids, sulfates and glycosides have been shown to be present in several species of *Aerva* [16,17,18,19]. Previous studies have reported that *Aerva javanica* extracts (prepared using methanol, hexane, ethyl acetate, chloroform and water) exhibit antibacterial activity against various gram-positive and gram-negative bacterial species (*E. coli*, *K. pneumoniae*, *S. aureus* and *S. epidermis*) including methicillin-resistant *S. aureus* (MRSA). Similarly, methanol, chloroform and hexane extracts prepared from the leaves and flowers of this plant exhibit relatively higher levels of antibacterial activities [18], which is mainly attributed to the presence of luteolin-*O*-glycoside, apigenin, tricin and kaempeferol [18]. Various extracts described above have also been shown to exhibit antifungal activity against a number of fungal species (*A. flavus*, *A. fumigatus*, *A. niger* and *F. solani*) [16] as well as exhibit anticancer activity against breast and prostate cancer cell lines [19,20]. Some of the common biologically active compounds that have been identified from extracts of *Aerva javanica* include kampeferols [21], quercetin [22], ecdysteroids [23], isorhamnetin glycosides [24] and β-sitosterol [25].

*Anastatica hierochuntica* and *Aerva javanica* are part of the traditional medicine in Qatar and have not been previously analyzed extensively. Accordingly, the present study focuses on the isolation and structural elucidation of some of the bioactive components of *Anastatica hierochuntica* and *Aerva javanica* grown in Qatar and evaluate their antioxidant, antibacterial and antifungal activities. 

## 2. Results

### 2.1. Phytochemical Analysis of Anastatica hierochuntica and Aerva javanica Grown in Qatar

Qualitative analysis of important phytochemicals in various extracts of *Anastatica hierochuntica* and *Aerva javanica* grown in Qatar showed the presence of a variety of phytochemicals and the results of these tests are summarized in Table 1.

*Anastatica hierochuntica*: Glycosides and terpenoids were found in all of the extracts except for the methanol extract, whereas anthroquinones were found in all of the extracts except the dichloromethane. It is worth noting that only phenols were found in all of the extracts and likely the flavones are free of the glycosidic moiety.

*Aerva javanica*: Saponins were found in all of the extracts except ethyl acetate. The presence of flavonoids was found in all of the extracts except ethanol and ethyl acetate. Only glycosides and phenols were present in all of the extracts.

### 2.2. Antibacterial Activity of Anastatica hierochuntica and Aerva javanica Extracts

The disc diffusion method was used to assess the antibacterial activity of the extracts [28]. Extracts were tested at 5, 10, 25, 50, 100 and 200 µg/mL versus ampicillin at 100 µg/mL as the control and the mean diameter of the inhibition zone (MDIZ) was estimated in millimeters as compared to positive controls. The activities were dependent on the concentration of the extract, with the highest activity for 200 µg/mL (Table 2). The antibacterial activity was determined using gram-negative *E. coli*, and gram-positive *B. subtilis* and *S. aureus* species. The control MDIZ values for *E. coli*, *B. subtilis* and *S. aureus* were 24.38 mm, 27.09 mm and 37.76 mm, respectively. The acetone extract of *Anastatica hierochuntica* showed 83% (MDIZ = 20.15 mm) antibacterial activity of control against gram-negative E. coli. The gram-positive bacterial species *B. subtilis* and *S. aureus* showed 75% (MDIZ = 20.31 mm) and 72% (MDIZ = 27.19 mm) antibacterial activities of control, respectively. In the case of *Aerva javanica*, the highest activity of 75% (MDIZ = 18.21 mm) of control against *E. coli* was observed for the hexane extract. The highest activity against *S. aureus* was 75% (MDIZ = 28.32 mm) of control for the ethyl acetate extract. The acetone extract of *Aerva javanica* showed 85% (MDIZ = 23.03 mm) of control activity against the *B. subtilis* gram-positive species.

A comparison of the antibacterial activity of *Anastatica hierochuntica* and *Aerva javanica* grown in different regions, as reported in previous studies, is shown in Table 3.

### 2.3. Antioxidant Activity of Anastatica hierochuntica and Aerva javanica Extracts

Free radical scavenging activity of the investigated extracts was assessed using a direct DPPH assay [26]. In this assay, DPPH (deep purple colored free radical) turns pale yellow when reduced by the antioxidants present in the sample. Various extracts displayed different antioxidant activities measured as a percent inhibition with reference to the known standard antioxidants—α-tocopherol and ascorbic acid (Figure 1 and Figure 2). For *Anastatica hierochuntica* (undiluted), there was 98% activity for the butanol extract and 97% for the ethanol extract. This was higher than both standards. When diluted 10- and 100-fold, it was noticed that the activity reduced and measured lower than the standards. *Aerva javanica* (undiluted) showed 98% activity for the butanol extract and 96% for ethanol the extract. This was higher than both standards. Similar to *Anastatica hierochuntica*, *Aerva javanica* (when diluted 10- and 100-fold) also showed reduced activity and measured lower than the standards.

For *Anastatica hierochuntica*, the antioxidant activity was found to increase in the following order: methanol, ethyl acetate, dichloromethane, hexane, acetone, ethanol and butanol. For *Aerva javanica*, the antioxidant activity was found to increase in the following order: methanol, ethyl acetate, butanol, dichloromethane, hexane, acetone and ethanol. The IC_50_ value for all the extracts were also determined and these values are presented in Table 4. The IC_50_ values obtained are the lowest (most active) for the ethanol extracts in both plants and the highest IC_50_ values were observed in the methanol extracts of both plants (Table 4).

### 2.4. Characterization of Secondary Metabolites Present in Extracts of Anastatica hierochuntica and Aerva javanica

*Anastatica hierochuntica*: The hexane extracts which showed significant or higher antioxidant and antibacterial activity were further fractionated by HPLC to isolate pure compounds. This process led to fractionation and isolation of at least 10 compounds. Of these 10 compounds, 3 compounds with large enough quantities were used for further characterization and identification. Spectroscopy techniques (FTIR and NMR) and library screening were used to elucidate the structures of the 3 compounds. The compounds identified were β-sitosterol, campesterol and methyl- 9-(4-(3,4-dihydroxy-1′-methyl-5′-oxocyclohexyl)-2-hydroxycyclohexyl) nonanoate. The other seven compounds have been saved for identification in the future.

*Aerva javanica*: The ethyl acetate and methanol extracts, which showed significant or higher antioxidant and antibacterial activity, were further fractionated by HPLC to isolate pure compounds. This process led to fractionation and isolation of at least 13 compounds from the 2 extracts. Of these 13 compounds, 5 compounds with large enough quantity from the ethyl acetate extracts and 1 compound from the methanol extracts were used for further characterization and identification of the compounds. Spectroscopy techniques (FTIR and NMR) were used to elucidate structures of the five compounds. The compounds identified from the ethyl acetate extracts were lupenone, betulinic acid and lupeol acetate (Figure 3; Appendix A). The compounds identified from the methanol extracts were persinoside A and B (Figure 4; Appendix A).

Further fractionation in other extracts of *Anastatica hierochuntica* and *Aerva javanica,* plus isolation of large enough quantities of remaining compounds for the hexane extracts of *Anastatica hierochuntica*, and ethyl acetate and methanol extracts of *Aerva javanica*, and determination of their biological properties of the compounds, is ongoing.

### 2.5. Antibacterial and Antifungal Activity of Isolated Compounds for Both the Plants

Antibacterial action of isolated compounds β-sitosterol, campesterol and methyl-9-(4-(3,4-dihydroxy-1′-methyl-5′-oxocyclohexyl)-2-hydroxycyclohexyl)nonanoate of *Anastatica hierochuntica* and lupenone, betulinic acid, lupeol acetate and persinoside A and B from *Aerva javanica* at 2000 g/mL were evaluated as compared to the control chloramphenicol 200 µg/mL. These data and the mean diameter inhibition zones (MDIZ) are shown in Table 5. The antifungal activity of the isolated compounds compared to Fluconazole 1250 µg/mL as control are shown in Table 6.

## 3. Materials and Methods

### 3.1. Materials

2,2-Diphenyl-1-picrylhydrazyl (DPPH) and organic solvents (acetone, ethanol, butanol, ethyl acetate, methanol, dichloromethane and hexane) were purchased from Sigma–Aldrich Chemical Co., St Louis, MO, USA. All other chemicals used were of analytical grade and available locally.

### 3.2. Methods

#### 3.2.1. Collection of Plants

*Anastatica hierochuntica* and *Aerva javanica* were collected from their natural habitats in the desert near Doha, Qatar. *Anastatica hierochuntica* were collected with the help of the elderly local population and a botanist from the roadside of Abu Samra Road, near Trainah, Southern Dukhan, Qatar (25°25′40.8144″ N and 50°46′59.9052″ E) and Al Jumayliyah of Western Qatar (25°36′34″ N, 51°5′32″ E). *Aerva javanica* were also collected in a similar manner using the knowledge of the local elderly population and a botanist from Salwa Road in South-Western Qatar—Al Luqta area (25.3107° N, 51.4651° E), Doha and in Jaow Al Hamar depression near Trainah, Southern Qatar (24°45′32″ N, 51°12′43″ E).

#### 3.2.2. Extraction of Phytochemicals and Fractionation

The aerial plant parts of *Anastatica hierochuntica* and *Aerva javanica* (twigs with bark, leaves and flowers) were ground into a fine powder, sieved (with particles smaller than 0.2 mm) and kept in sealed plastic bags in a clean and dry cabinet. Approximately 50 g of this material was extracted with various organic solvents by the Soxhlet solvent extraction method. For the Soxhlet solvent extraction, the plant material was wrapped in Whatman #1 filter paper and inserted into the Soxhlet column. The temperature of the heating mantle holding the round bottom flask was set slightly above the boing point of the solvent, and 100 mL of the solvent was allowed to percolate through the Soxhlet for at least 10 cycles. The typical extraction times ranged between 10 and 12 h. The solvents used were **a**: acetone; **b**: butanol; **c**: ethanol; **d**: ethyl acetate; **e**: methanol; **f**: dichloromethane; **g**: hexane. The extracts were concentrated to approximately 10 mg/mL using a Rotovap (Rota vapor R210, Buchi, Flawil, Switzerland) and stored at −20 °C until further use.

#### 3.2.3. Phytochemical Analysis

The phytochemical analysis of the extracts for the identification of alkaloids, glycosides, tannins, flavonoids, terpenoids, saponins, etc., were carried out using previously reported procedures [26,27]. The extracts that contained the highest levels of phytochemicals were then analyzed for isolation of the biologically active compounds.

#### 3.2.4. Antibacterial Activity Tests

The disc diffusion method was employed to evaluate the antibacterial activity. Ampicillin (100 µg/mL) or chloramphenicol (200 µg/mL) was selected as the positive control. Extracts were tested at 5, 10, 25, 50, 100 and 200 µg/mL versus ampicillin (100 µg/mL) or versus chloramphenicol (200 µg/mL) as the control and the diameter of zone inhibition was estimated in millimeters as compared to positive controls. One-day-old stock bacterial cultures (25 μL) with an absorbance of 1.2 optical density units at 600 nm was used to spread over LB agar plates. After allowing the culture to soak up for 15 min, the plate was divided into four quadrants; a wick of filter paper was placed in the middle of each quadrant. The positive control and test compounds were added to the filter paper wicks. Additional filter paper wicks were used to catch any overflow of the solution. The plates were incubated at 37 ºC for 24 h, and then observed for any clear zone around the filters [28]. In case of positive tests, the clearing zone were calculated by measuring their radius and then converting to the diameter. The bacterial species used in this study were obtained from the American Type Culture Collection and have been propagated in our laboratory over the years.

#### 3.2.5. DPPH In Vitro Assay

The concentration of the extracts was adjusted to 10 mg/mL and their antioxidant activities were tested using DPPH (2,2-diphenyl-1-picrylhydrazyl) at 1-, 10- and 100-fold dilutions. These test results were compared to the antioxidant activity of α-tocopherol (1 mg/mL) and ascorbic acid (1 mg/mL) which served as the positive control. The extracts were mixed with DPPH (1 mL of 0.1 mM solution in methanol) in a test tube and shaken vigorously before being left in the dark at 27 °C for 45 min. The control sample was prepared in the same way but without the extracts. The absorbance of the tested samples was measured by a spectrophotometer at 517 nm. The total antioxidant activity of the tested extracts/samples was determined as an inhibition percentage, calculated using the formula below [35]:

DPPH scavenging effect (%) or percent inhibition:
$$\frac{\mathrm{Absorbance}\;\mathrm{of}\;\mathrm{control}-\mathrm{Absorbance}\;\mathrm{of}\;\mathrm{sample}}{\mathrm{Absorbance}\;\mathrm{of}\;\mathrm{control}}\times 100$$

#### 3.2.6. Identification and Characterization of the Active Compounds

For the isolation of active compounds, both *Anastatica hierochuntica* and *Aerva javanica* plant extracts/fractions were analyzed using the preparative HPTLC and HPLC. The HPTLC was conducted with a CAMAG^®^ HPTLC scanner III, using hexane and ethyl acetate in the ratio of 7:3 as the solvent. The HPLC was completed on a Shimadzu SPD 20A HPLC instrument with a C18 column (4.6 mm × 250 mm; 0.5 micron), acetonitrile and water (15:85), with 0.1% phosphoric acid as the solvent at a flow rate of 1 mL/min. The UV spectra were then recorded on a PDA detector. The compounds were further characterized with IR and NMR spectroscopic techniques. The NMR spectra were recorded using a Bruker 400 MHz spectrometer for ^1^H and ^13^C using deuterated chloroform (CDCl_3_). The chemical shifts were expressed in δ (ppm) down-field from the tetramethylsilane as the internal standard. The IR spectra were recorded by a Bruker FT-IR spectrophotometer from 4000 to 500 cm^−1^.

#### 3.2.7. Data Analysis

The Microsoft Excel computer program (Microsoft, Inc., Redmond, WA, USA) was used to generate concentration vs. % inhibition plots to determine IC_50_ values as well as mean and standard deviations.

## 4. Discussion

Qatar has very limited vegetation available due to its heavily weathered soil and prevailing climatic conditions. *Anastatica hierochuntica* and *Aerva javanica* are among the very few plants that grow in this area of the Arabian Peninsula and are used as part of the traditional medicine by the local population. This provides scope for investigating these plants for their potential therapeutic activity. Furthermore, the natural products obtained from these plants, either as pure compounds or as extracts, can provide opportunities for new drug leads. Due to an increasing demand for chemical diversity in screening programs, search for therapeutic drugs from the natural vegetation present locally are in huge demand.

### 4.1. Qualitative Phytochemical Screening of Anastatica hierochuntica and Aerva javanica Extracts

The qualitative phytochemical screening carried out on various extracts of *Anastatica hierochuntica* and *Aerva javanica* showed the presence of secondary metabolites, i.e., alkaloids, glycosides, tannins, flavonoids, terpenoids, saponins, phenols and anthraquinones. Summarized below (Table 7) are the biological activities of the aforesaid secondary metabolites.

The medicinal value of both the plants among the indigenous population of Qatar is due to the presence of the above-mentioned biologically active and therapeutically effective phytochemicals present in them. It is noteworthy to mention that in our study, the anthraquinone test performed on both *Anastatica hierochuntica* and *Aerva javanica* produced a unique greenish-yellow precipitate instead of the expected pink-violet or red coloration. This observation possibly suggests the presence of a modified or new anthraquinone compounds. The analysis further revealed the absence of proteins, anthocyanins and phlobatannins in all the solvent extracts for both *Anastatica hierochuntica* and *Aerva javanica*.

### 4.2. Antibacterial Activity of Anastatica hierochuntica and Aerva javanica Extracts

A few earlier studies have evaluated and established a wide range of antibacterial activities for various extracts of Anastatica hierochuntica [9,12,13]. In our study, we used the reference bacterial strains gram-negative *E. coli*, and gram-positive *B. subtilis* and *S. aureus*. These were tested using various extracts of *Anastatica hierochuntica* and *Aerva javanica*. It was observed that the acetone extract of the plant showed the highest antibacterial activity against *E. coli*. The hexane extract showed the highest activity against *S. aureus*, whereas significant activity was observed for ethyl acetate extract against *B. subtilis*. A study conducted earlier in Egypt has shown no antibacterial activity against *E. coli* and *S. aureus* in methanol extracts, although a strong antibacterial activity against *B. subtilis* in the methanol extract was observed [30,31]. However, in our investigation, we found a contradicting observation of low antibacterial activity for *E. coli*, *S. aureus* and *B. subtilis* in the methanol extracts of these plants. This could be attributed to the stressful environment in which these plants grow in Qatar and the relatively higher concentrations of secondary metabolites found in these plants.

Similarly, there are several studies documenting a wide range of antibacterial activity against clinically significant gram-negative and gram-positive bacterial strains for various extracts of *Aerva javanica* [18,22,23]. In our study, antibacterial activity was conducted against gram-negative *E. coli*, and gram-positive *S. aureus* and *B. subtilis*. Significant antibacterial activity for *E. coli* was shown in the hexane extract, while for *S. aureus* the highest antibacterial activity was observed in the acetone extract. Ethyl acetate showed the highest antibacterial activity for *B. subtilis*. The results obtained by our group for *Aerva javanica* grown in Qatar exhibits a contrasting observation from previously reported studies on *Aerva javanica* from different regions by other investigators [18,23,24]. For example, in a study conducted in the Albaha region, Saudi Arabia, the antibacterial activity of dichloromethane, ethanol and methanol extracts of different parts of *Aerva javanica* revealed that *E. coli* growth was inhibited only by dichloromethane extracts. *S. aureus* growth was inhibited by only the methanol extract of the leaves. All of the extracts had zero inhibitory effect on *B. subtilis* [24]. Another study conducted in Jazan, Saudi Arabia, established that *Aerva javanica* had an antibacterial effect on *E. coli* species for its ethanol extract and *S. aureus* growth was inhibited by all of the solvent extracts (see above) that were investigated [23]. Further, a study conducted in India identified antibacterial activity against *E. coli*, *S. aureus* and *B. subtilis* in hexane, chloroform and methanol extracts of *Aerva juvanica*, with the highest activity displayed by the methanolic extracts [18].

The current findings for both *Anastatica hierochuntica* and *Aerva javanica* grown in Qatar suggests that the desert environment of Qatar can have a profound effect on the type and number of active compounds present. This can be attributed to low levels of water, soil infertility, intense sunlight and heat. Since these conditions are difficult for the plants to survive, they produce a higher concentration of secondary metabolites. Further, the difference in antibacterial activities may be attributed to several factors that include: (i) solubility of bioactive components in different extracts [40]; (ii) part of the plant used in the study; (iii) isolation process used [41]; (iv) geographical area and varying climatic conditions. The conditions listed above are likely to alter the phytochemical composition of the plants, and accordingly, the variation in antibacterial activity [42,43]. The presence of various phytochemical constituents such as alkaloids, flavonoids, terpenoids, tannins, saponins and some phenolic compounds were responsible for the antibacterial activity of the plant extracts [43].

### 4.3. Antioxidant Activity of Anastatica hierochuntica and Aerva javanica Extracts

The antioxidant activity of different extracts of *Anastatica hierochuntica* were measured via the widely used 2,2-diphenyl-1-picrylhydrazyl (DPPH) radical scavenging assay. Ascorbic acid and α-tocopherol were used as the standards. Our studies show that various extracts of both the plants exhibit significant antioxidant activities. The DPPH assay shows 98% inhibition in butanol and 97% in ethanol-undiluted extracts, which was higher when compared with both the standards used. Similarly, significant antioxidant activity was also observed in *Aerva javanica* extracts as well. The DPPH assay displays 98% inhibition in butanol and 97% in ethanol-undiluted extracts. These values were higher than both the standards used. On dilution of the extracts to 10- and 100-fold, both *Anastatica hierochuntica* and *Aerva javanica* extracts showed reduced antioxidant activities and these values were lower than the standards used for analysis. The IC_50_ values determined further confirm the above strong antioxidant activities in each of the plants. As per IC_50_ values, the ethanol extracts show the strongest antioxidant activity, and the methanol extracts show the lowest antioxidant activity. Several researchers have stated the significance of the phytochemical constituents—phenols, terpenes, flavonoids, tannins, saponins and sterols— in assisting free radical scavenging, thereby enhancing the antioxidant activity [44]. It is observed that all of these listed phytochemicals are present in *Anastatica hierochuntica* and *Aerva javanica* and might be responsible for the antioxidant properties exhibited by both of these plant extracts. These compounds are known to neutralize the effect of free radicals and inhibit the oxidation of cell membranes and other molecules. In addition, in *Anastatica hierochuntica*, we have characterized and isolated phytosterols, viz., campesterol and β-sitosterol, which are well known to exhibit antioxidant activity [45,46]. Further, in this study, we have also identified triterpenoids—lupenone, lupeol acetate and betulinic acid—in the extracts of *Aerva javanica* and these compounds have been reported previously to be potent antioxidants [47,48,49]. Even though our studies have shown significant antioxidant properties for the plant extracts, more extensive studies and tests need to be conducted to further confirm the use of extracts as antioxidants.

### 4.4. Compounds Isolated from Anastatica hierochuntica and Aerva javanica Extracts and Their Role in Enhancing Biological Activities

In this investigation, various extracts of *Anastatica hierochuntica* and *Aerva javanica* have been further subjected to reverse phase CombiFlash chromatography and HPLC to isolate pure compounds and characterize them using IR and NMR spectral analysis methods. Thus, isolated compounds from the extracts of *Anastatica hierochuntica* include β-sitosterol, campesterol and methyl-9-(4-(3,4-dihydroxy-1′-methyl-5′-oxocyclohexyl)-2-hydroxycyclohexyl)nonanoate. Similarly, from the extracts of *Aerva javanica*, the isolated and characterized pure compounds include lupenone, lupeol acetate, betulinic acid, and persinoside A and B.

The phytosterol, β-sitosterol, that we identified in the hexane extract of *Anastatica hierochuntica* has been previously reported from the plants *Malva parviflora* [33], *Senecio lyratus* [50] and *Parthenium hysterophorus* [51]. Studies on *Malva parviflora* [33] revealed that the ethanol and chloroform extracts of the plant indicates the presence of β-sitosterol as an active component that is capable of inhibiting the growth of bacterial species *S. aureus* and *E. coli*. Investigations reported by Kiprono and associates [50] on *Senecio lyratus* established that β-sitosterol exhibits both antibacterial and antifungal activity. Similarly, β-sitosterol isolated from *Parthenium hysterophorus* has been reported to have promising antibacterial activity and is widely used in aquaculture to inhibit the growth of vibrio infection [51]. In addition, another study demonstrated that β-sitosterol inhibits the growth of various microorganisms with a zone of inhibition ranging from 15 to 28 mm. In the latter study, the activity of β-sitosterol was found to be weaker compared with the standard antibacterial agent, gentamycin, and the standard antifungal agent, ticonazole [52]. In our research, β-sitosterol had much weaker inhibition zones of 4 mm when applied at 2000 µg/mL, but was found to have antifungal activity, resulting in a 20 mm diameter of inhibition zone. Additionally, β-sitosterol has been reported in the literature to exhibit a variety of pharmacological activities such as anti-inflammatory, anticancer, antioxidant, chemo protectant, antidiabetic and anorexia activity [53,54,55]. Although consistent with the above study [52], the weaker antibacterial activity observed could be because of ineffective diffusion of the compounds in aqueous solutions at the concentrations used. On the other hand, the precipitation of the compounds on the filter paper discs and/or around it was not observed. Although unlikely, slight variations in the methods of analysis used could also add to this observation.

Campesterol, another phytosterol isolated from the hexane extract has also been extensively reported from various plants and its activity has also been studied to a commendable extent. Its structure has a cholesterol skeleton with a different side chain and shows cholesterol-lowering effects. Campesterol is also found to have antibacterial, anti-inflammatory, antifungal and anticancer activities [56]. The anti-inflammatory property exhibited by campesterol explains the traditional use of *Anastatica hierochuntica* to treat menstrual cramps [4]. The other compound, methyl-9-(4-(3,4-dihydroxy-1′-methyl-5′-oxocyclohexyl)-2-hydroxycyclohexyl) nonanoate identified in the hexane extract of *Anastatica hierochuntica* has not been reported on previously. Preliminary investigations have revealed that this compound exhibits both weak antibacterial activity against *E. coli* and *S. aureus* and weak anti-fungal activity against *Saccharomyces cerevisiae* and *A. fumigatus*. Further detailed research for biological activities of this compound is underway.

Lupenone and lupeol acetate (triterpenoid) that we have isolated from the ethyl acetate extract of *Aerva javanica* has also been reported previously. One of the recent reports on the plant *Himanthus drasticus* showed a diverse array of biological activities such as anti-inflammatory, antibacterial and antioxidant capabilities for lupenone and lupeol acetate [57]. A study published on the bioactivity of triterpenoids isolated from *Euphorbia Segetalis* reported that lupenone exhibits strong antiviral activity against the herpes simplex virus [58]. Some reports have also shown that lupeol acetate (100 µg/mL) inhibits the growth of various microorganisms with a zone of inhibition ranging from 11 to 18 mm. This is considered weaker antibacterial activity when compared to the standard antibacterial agent, ciprofloxacin [34]. Furthermore, in our study, lupeol acetate exhibited a much weaker zone of inhibition (2–4 mm) for antibacterial and antifungal (4–6 mm) activities at 2000 µg/mL.

Betulinic acid, a lupane-type pentacyclic triterpene derived from *Tetracera potatoria*, has been reported to be a potent analgesic, anti-inflammatory, and antipyretic agent [59]. In recent years, the significant antitumor activity of betulinic acid has been reported by several research groups [60,61]. In addition, betulinic acid has been found to be a promising candidate as an anti-inflammatory and anti-HIV agent [62]. Persinoside A and B flavanone glucosides have been isolated and reported earlier from *Aerva persica* and exhibit antioxidant activities [63]. Additionally, in our research, betulinic acid showed a diameter of zone of inhibition 2–4 mm, and antifungal activity of 6–10 mm at concentrations of 2000 μg/mL. Although weak, these values, again, are consistent with the observations of others.

## 5. Conclusions

In conclusion, secondary metabolites from plant extracts can possess a wide range of biological activities and thus have potential therapeutic uses. They can also be a rich source of lead compounds that can then be structurally and chemically modified in drug design for the development of new drugs. This study represents the continuity of work from our lab to characterize locally available plants that grow in the desert environment of Qatar. Data obtained from the current investigation confirmed the presence of biologically active secondary metabolites in both the plants, viz., *Anastatica hierochuntica* and *Aerva javanica*. The extracts of these plants had potential antioxidant activity and antibacterial activity against *E. coli.*, *S. aureus* and *B. Subtilis*. In addition, the compounds isolated from both of these plants, *Anastatica hierochuntica* and *Aerva javanica*, are known to have a wide range of medicinal capabilities. Further, we have isolated and characterized β-sitosterol, campesterol and methyl-9-(4-(3,4-dihydroxy-1′-methyl-5′-oxocyclohexyl)-2-hydroxycyclohexyl)nonanoate from the extracts of *Anastatica hierochuntica*, and triterpenoids (lupenone, lupeol acetate, betulinic acid) and flavonoids (persinoside A and B) from the extracts of *Aerva javanica*. The purified compounds have also been tested for their antibacterial and antifungal activities. The therapeutic properties of some of these isolated compounds is worth mentioning. Studies have indicated that β-sitosterol has a wide range of pharmacological effects, including anti-inflammatory, anticancer, antioxidant, chemoprotective, antidiabetic and anorectic properties, whereas campesterol is also found to have antibacterial, anti-inflammatory, antifungal and anticancer activities. Betulinic acid is a potent analgesic, anti-inflammatory, antitumor, anti-HIV and antipyretic agent. Therefore, if explored further, these medicinal plants used by the indigenous population of Qatar can not only be used as good alternative for synthetic drugs but also as a promising source for designing bioactive compounds.

## Figures and Tables

**Figure 1 molecules-28-03364-f001:**
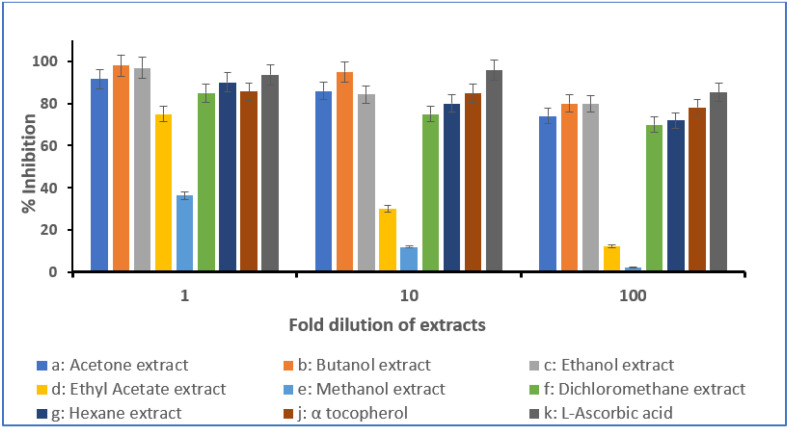
Antioxidant activity of *Anastatica hierochuntica* extracts by DPPH method. Various extracts prepared from *Anastatica hierochuntica* were subjected to the DPPH assay as described in the Section 3. The inhibition activity was compared to known antioxidants (α-tocopherol and ascorbic acid).

**Figure 2 molecules-28-03364-f002:**
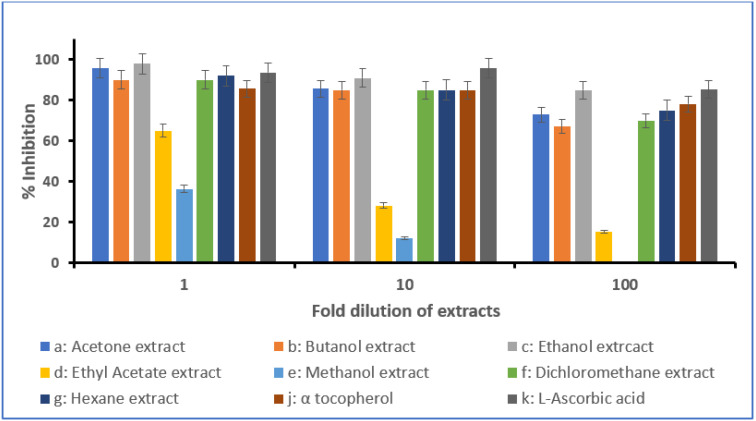
Antioxidant activity of the *Aerva javanica* extracts by the DPPH method. Various extracts prepared from *Aerva javanica* were subjected to the DPPH assay as described in the Section 3. The inhibition activity was compared to known antioxidants (α-tocopherol and ascorbic acid).

**Figure 3 molecules-28-03364-f003:**
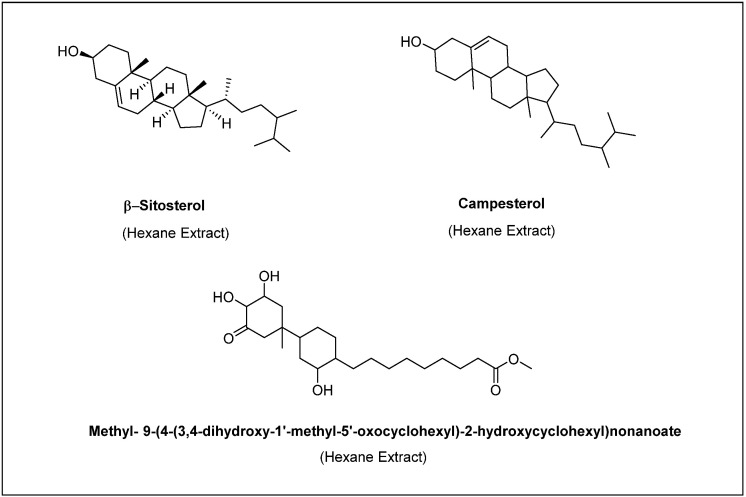
The major secondary metabolites identified in the hexane extracts of *Anastatica hierochuntica*.

**Figure 4 molecules-28-03364-f004:**
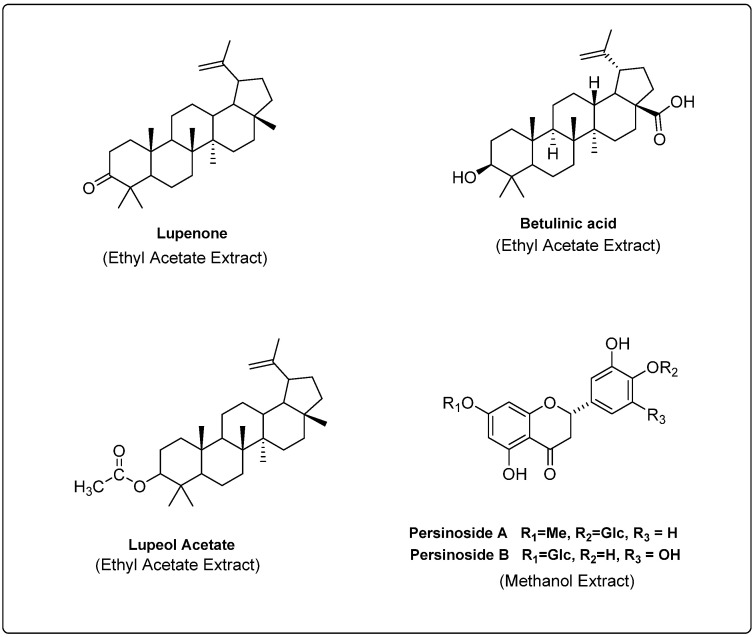
The major secondary metabolites identified in the ethyl acetate and methanol extracts of *Aerva javanica*.

**Table 1 molecules-28-03364-t001:** Phytochemical analysis of *Anastatica hierochuntica* and *Aerva javanica* grown in Qatar.

No.	Test	Procedure *	Observation	*Anastatica hierachuntica*							*Aerva javanica*						
				Extracts *							Extracts *						
				a	b	c	d	e	f	g	a	b	c	d	e	f	g
1	Alkaloids	Wagner’s reagent	Brown/reddish ppt.	+	−	−	−	−	+	+	+	−	−	−	−	+	+
2	Glycosides	Keller–Kiliani reagent	Brown ring at the junction	+	+	+	+	−	+	+	+	+	+	+	+	+	+
3	Tannins	Braymer’s test	Greenish ppt.	+	−	+	+	+	+	+	−	+	+	−	+	+	+
4	Flavonoids	Drop of 10% NaOH solution	Intense yellow color	−	+	+	+	+	+	+	+	+	−	−	+	+	+
5	Terpenoids	Salkowski test	Reddish brown ring at the junction	+	+	+	+	−	+	+	+	+	+	−	−	+	+
6	Saponins	Foam test	Stable froth produced	+	−	−	+	+	+	+	+	+	+	−	+	+	+
7	Phenol	2 mL 5% FeCl_3_	Blue-green	+	+	+	+	+	+	+	+	+	+	+	+	+	+
8	Anthraquinones	Benzene + 26% NH_3_	Greenish-yellow precipitate ^#^	+	+	+	+	+	−	+	+	+	+	−	+	−	+
9	Phlobatannins	2 mL 1% HCl + heat	Red ppt.	−	−	−	−	−	−	−	−	−	−	−	−	−	−
10	Anthocyanins	2 mL HCl (2M) + ammonia (NH_3_)	No change	−	−	−	−	−	−	−	−	−	−	−	−	−	−
11	Proteins	Xanthoproteic test	No change	−	−	−	−	−	−	−	−	−	−	−	−	−	−

* The qualitative tests used for phytochemical analysis were as described previously [26,27]; +: Positive test; −: Negative test; a: acetone; b: butanol; c: ethanol; d: ethyl acetate; e: methanol; f: dichloromethane; g: hexane. ^#^ The anthraquinones were expected to produce a pink, violet or red coloration in the ammonical layer, but a greenish-yellow precipitate formation was observed.

**Table 2 molecules-28-03364-t002:** Antibacterial activity of *Anastatica hierochuntica* and *Aerva javanica* extracts grown in Qatar.

Bacterial Species Tested	Extract *	MDIZ, mm (% of Control)	
		*Anastatica hierochuntica*	*Aerva javanica*
*Escherichia coli*	a	20.15 (83)	6.80 (28)
	b	0	0
	c	18.21 (75)	7.28 (30)
	d	14.57 (60)	8.50 (35)
	e	3.16 (13)	6.07 (25)
	f	16.70 (70)	17.00 (70)
	g	19.42 (80)	18.21 (75)
*Bacillus subtilis*	a	20.31 (75)	23.03 (85)
	b	17.61 (65)	8.13 (30)
	c	7.86 (29)	6.77 (25)
	d	0	0
	e	0	0
	f	17.61 (65)	13.55 (50)
	g	20.31 (75)	17.61 (65)
*Staphylococcus aureus*	a	27.19 (72)	20.75 (55)
	b	0	24.17 (64)
	c	24.54 (65)	18.12 (48)
	d	28.32 (75)	27.19 (72)
	e	9.44 (25)	13.22 (35)
	f	25.30 (67)	22.66 (60)
	g	26.43 (70)	28.32 (75)

* a: acetone; b: butanol; c: ethanol; d: ethyl acetate; e: methanol; f: dichloromethane; g: hexane. Ampicillin (100 µg/mL) was used as the standard/positive control. The control MDIZ values for *E. coli*, *B. subtilis* and *S. aureus* were 24.38 mm, 27.09 mm and 37.76 mm, respectively. The concentration of the plant extracts used for generating the data reported above is 200 µg/mL. Reference value: active ≡ MDIZ ≥ 14 mm, moderately active ≡ MDIZ = 10–13 mm, inactive ≡ MDIZ < 10 mm [29].

**Table 3 molecules-28-03364-t003:** Antibacterial activities of various extracts of *Anastatica hierochuntica* and *Aerva javanica* growing in different regions: A comparison.

Plant	Plant Source	Extract	Organism			Part of Plant Tested	Ref
			*E. coli*	*B. subtilis*	*S. aureus*		
*Anastatica hierochuntica*	Qatar	Acetone	+	+	+	Stem, leaves Concentrations used: 5–200 µg/mL	Present Study
		Butanol	−	+	-		
		Ethanol	+	+	+		
		Ethyl Acetate	+	−	+		
		Methanol	+	−	+		
		Dichloromethane	+	+	+		
		Hexane	+	+	+		
	Egypt	Methanol	−	+	-	Whole Plant	[9]
	Iraq	Methanol	+	−	+	Whole Plant	[30]
	Algeria	Methanol	+	−	+	Whole Plant	[31]
	Saudi Arabia	Methanol	+	+	+	Whole Plant	[32]
*Aerva javanica*	Qatar	Acetone	+	+	+	Stem, leaves Concentrations used: 5–200 µg/mL	Present Study
		Butanol	−	+	-		
		Ethanol	+	+	+		
		Ethyl Acetate	+	−	+		
		Methanol	+	−	+		
		Dichloromethane	+	+	+		
		Hexane	+	+	+		
	India	Methanol	+	+	+	Leaf, Flower, Root, Stem	[18]
		Hexane	+	+	+		
	Saudi Arabia	Ethanol	+	ND	+	Root, Leaves [21]	
		Methanol	ND	ND	+		
		Petroleum Ether *	ND	ND	+		
		Acetone	ND	ND	+		[24]
Saudi Arabia		Methanol	+	−	+		
		Dichloromethane	+	−	−		

ND = not determined. * Petroleum ether comparable to hexane. +: Positive test; −: Negative test.

**Table 4 molecules-28-03364-t004:** Antioxidant activities of *Anastatica hierochuntica* and *Aerva javanica* IC_50_ values (mg/mL).

Plant Extract	IC_50_ Values (mg/mL) *	
	*Anastatica hierochuntica*	*Aerva javanica*
Acetone	0.022 ± 0.001	0.030 ± 0.002
Butanol	0.023 ± 0.001	0.040 ± 0.002
Ethanol	0.016 ± 0.001	0.013 ± 0.001
Ethyl Acetate	2.18 ± 0.110	1.92 ± 0.096
Methanol	3.00 ± 0.150	2.82 ± 0.141
Dichloromethane	0.017 ± 0.001	0.027 ± 0.001
Hexane	0.021 ± 0.001	0.020 ± 0.001

* Values are mean ± SD of three determinations.

**Table 5 molecules-28-03364-t005:** Antibacterial activity of metabolites identified in extracts of *Anastatica hierochuntica* and *Aerva javanica*.

Plant	Isolated Compound	Organism	Results *	Literature Report *
*Anastatica hierochuntica*	β-Sitosterol	*E. coli*	4 mm	~15 mm [33]
	Campesterol		4 mm	N/A
	Methyl-9-(4-(3,4-dihydroxy-1′methyl-5′-oxocyclohexyl)-2-hydroxycyclohexyl)nonanoate		4 mm	N/A
	β-Sitosterol	*S. aureus*	2 mm	~18 mm [33]
	Campesterol		2 mm	N/A
	Methyl-9-(4-(3,4-dihydroxy-1′methyl-5′-oxocyclohexyl)-2-hydroxycyclohexyl)nonanoate		2 mm	N/A
*Aerva javanica*	Lupenone	*E. coli*	4 mm	N/A
	Betulinic acid		4 mm	N/A
	Lupeol acetate		4 mm	Inactive [34]
	Persinoside A and B.		4 mm	N/A
	Lupenone	*S. aureus*	6 mm	N/A
	Betulinic acid		2 mm	N/A
	Lupeol acetate		2 mm	Inactive [34]
	Persinoside A and B.		2 mm	N/A

* Reference value: active ≡ MDIZ ≥ 14 mm, moderately active ≡ MDIZ = 10–13 mm, inactive ≡ MDIZ < 10 mm [29]; ~ = Approximately; N/A = Not available.

**Table 6 molecules-28-03364-t006:** Antifungal activity of metabolites identified in acetone extracts of *Anastatica hierochuntica* and *Aerva javanica*.

Plant	Isolated Compound	Organism	Results *
*Anastatica hierochuntica*	β-Sitosterol	*Saccharomyces cerevisiae*	20 mm
	Campesterol		6 mm
	Methyl-9-(4-(3,4-dihydroxy-1′methyl-5′-oxocyclohexyl)-2-hydroxycyclohexyl)nonanoate		10 mm
	β-Sitosterol	*A. fumigatus*	20 mm
	Campesterol		10 mm
	Methyl-9-(4-(3,4-dihydroxy-1′methyl-5′-oxocyclohexyl)-2-hydroxycyclohexyl)nonanoate		6 mm
*Aerva javanica*	Lupenone	*Saccharomyces cerevisiae*	4 mm
	Betulinic acid		6 mm
	Lupeol acetate		6 mm
	Persinoside A and B.		6 mm
	Lupenone	*A. fumigatus*	4 mm
	Betulinic acid		10 mm
	Lupeol acetate		4 mm
	Persinoside A and B.		2 mm

* Reference value: active ≡ MDIZ ≥ 14 mm, moderately active ≡ MDIZ = 10–13 mm, inactive ≡ MDIZ < 10 mm [29].

**Table 7 molecules-28-03364-t007:** Biological activities of various secondary metabolites: Summary.

Secondary Metabolite	Reported Biological Activity *	Reference
Alkaloids	i, ii, iii, iv, v, vi, vii, viii and ix	[36]
Tannins	i, ii, iii, v, vi, x, and xi	[37]
Flavonoids	i, ii, iii, iv, v, xii and xiii	[38]
Terpenoids	i, iii, xiv, xv and xvi	[36]
saponins	i, iv, xvii, xviii, and xix	[39]
Phenols	i, ii, iii, vi, v, xii and xiii	[38]

* Reported biological activities i–xix are as follows. i: antibacterial; ii: antioxidant; iii: antiinflammatory; iv: anticancer; v: cardioprotective; vi: antidiabetic; vii: sedatives; viii: neuroprotective; ix: insecticidal; x: antinociceptive; xi: antipyretic; xii: immunomodulatory; xiii: protection against UV radiation; xiv: anti-rheumatic; xv: antimalarial; xvi: hepatocidal; xvii: antifungal; xviii: antiparasitic; xix: cholesterol lowering effects.

## Data Availability

Not applicable.

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
