# Peer review of "Phytochemical Analysis of Anastatica hierochuntica and Aerva javanica Grown in Qatar: Their Biological Activities and Identification of Some Active Ingredients"

_molecules, 2023, doi:10.3390/molecules28083364_

Round 1
Reviewer 1 Report
In this work, the authors conducted a lot of experiments to analyze the biological activities and active constituents in Anastatica hierochuntica and Aerva javanica grown in Qatar. However, before accepted, a major revision is needed for the manuscript.
1. Anastatica hierochuntica and Aerva javanica belong to different families. Taken them as materials in this research, just because they have same effects?
2. In fact, so much study was done on Anastatica hierochuntica and Aerva javanica, including the antioxidant activities, antibacterial and antifungal activities, even their chemical composition. And what is the authors’ purpose to do the research?
3. The Latin names of these two plants begin with "A", but they belong to different families and genera. Please write the full names instead of abbreviations.
4. The research need to be optimized.
5. Pay more attention to the logic of language. For example, In Line 96, “It is a perennial and an indigenous drought tolerant herb widespread in central and southern parts of Qatar. It also grows in Asia, Africa, Arabian peninsula Egypt, and many other parts of the world.” Qatar is not an Asian Country?
Reviewer 2 Report
In the present manuscript entitled, "Phytochemical Analysis of Anastatica hierochuntica and Aerva javanica Grown in Qatar: Their Biological Activities and Identification of Active Ingredients," the authors have performed the phytochemical analysis, and biological activities of various solvent extracts of Anastatica hierochuntica and Aerva javanica. Identification of phytocompounds in active extract was performed using HPLC, NMR technique. Further activities were performed with isolated compounds to validate whether these compounds are responsible for their activity or not. Overall, the study was designed and performed nicely with good presentation and discussion with previous literature.
However, there are few minor corrections required in the present manuscript as listed below-
1. According to journal instructions, abstract should be of 200 words maximum.
2. There was plagiarism in Method section especially in 2.2.2, 2.2.3, 2.2.4, 2.2.5 sections.
3. The antioxidant activity can be expressed as IC50 values, if possible, please compare your data by calculating IC50 of each extract.
Reviewer 3 Report
The manuscript “Phytochemical Analysis of Anastatica hierochuntica and Aerva javanica Grown in Qatar: Their Biological Activities and Identification of Active Ingredients" is devoted to a comprehensive study of the antimicrobial activity of various extracts of two plants grown in the desert environment of Qatar. Topic of this work is quite relevant. Antibacterial activity against Escherichia coli, Bacillus subtilis, and Staphylococcus aureus was estimate. Antioxidant activity was estimated using DPPH method. It should be noted that the advantage of this work is the identification of individual components and the establishment of their structure. Despite the fact that the results of the work contribute to the expansion of existing data, the main recommendation to the authors is to specify the originality of the work. Undoubtedly, the results of this work can be useful for biochemistry and pharmacy.
The manuscript is written in a good language, has a good structure and a clear representation of the data. I think, this manuscript can be published in the Molecules journal after minor revision after taking into account comments given below:
- Materials: the coordinate of plants collection should be specified.
- Methods: Soxhlet extraction conditions should be revealed (number of cycles, control of extraction during the process, mass/volume ratio itc.).
- Phytochemical analysis: methods should be briefly described.
- Antibacterial activity tests: How was the density of bacterial suspension determined? How was the identity of bacteria confirmed?
- Provide the statistical analysis of the results.
- Table 3 is partially missed.
- Table 6 looks badly formatted.
Reviewer 4 Report
The manuscript entitled “Phytochemical Analysis of Anastatica hierochuntica and Aerva javanica Grown in Qatar: Their Biological Activities and Identification of Active Ingredients” need major revisions before further consideration for publication. However, this manuscript contains valuable information that needs to be re-organized, and some experiments should be done in order to give more strength to the results. For example, the determination of IC50 in DPPH assay, The presentation of antibacterial zone of inhibition in mm. In addition, the following points should be addressed:
Abstract:
The abstract need to be more focused and concise. For example, the concentrations used for each biological activity is not indicated. The selection of extracts used for the isolation should be justified. Try to give only the relevant information in the abstract, this will help to reduce the word counts of the abstract.
Introduction
Line 69: correct “hepatoprotective” to “hepatoprotective”
After full description of A. hierochuntica and A. javanica which showed that several preliminary studies have been done so far on these plants, the authors did not present the novelty of the current study.
Material and methods
How were the plants (A. hierochuntica and A. javanica) identified? Please indicate the voucher specimen numbers after authentication by a plant taxonomist or botanist?
Antibacterial tests: The concentration of test extracts or compounds are not indicated.
DPPH assay: At least for the most active extracts, the determination of the IC50 should be done.
3.2. Antibacterial activity of Anastatica hierochuntica and Aerva javanica extracts
The area of inhibition or zone of inhibition is expressed in mm, and not mm2. The authors should present their results in mm. or what is the meaning of percentage of activity in this assay?
Table 3 should be rearranged.
Tables 4 and 5: Instead for the authors to give their results as “inactive”, the authors should present their results in mm, and the term “inactive” will just be the conclusion based on the reference values.
Conclusions:
The authors should be able to link the presence of isolated compounds with each biological assay. If not, the authors should indicate the limitations and perspectives of their study.
Round 2
Reviewer 1 Report
Accept as it is.
Author Response
Authors Response: Thank you for your suggestion. We have meticulously spell checked the whole manuscript and made some grammatical error correction as well.
Reviewer 4 Report
The major comments raised on this manuscript has been partially addressed. These following points are still not well explained:
1- The relationship between the presence of isolated compounds with the antibacterial activity. For example, the antibacterial activity of extracts are expressed in percentage, and for the isolated compounds, this activity is expressed in mm. So, how do you link these results? In addition, what is the meaning of percentage of activity in antibacterial assay? Moreover, the isolated compounds are inactive in antibacterial assay when compared with studies of other authors, a clear explanation should be given. Is it because of the solubility in the solvent used? or... ?
2- For the DPPH assay, the calculation of the IC50 can be done based on the data presented by the authors. In fact, the authors tested the extracts at three dilution factors, and the determination of IC50 can be done by grouping the data of each extract in a dose-dependent manner. Then by using an appropriate software for example GraphPad Prism, it will be possible to get the IC50s.
3- There are several grammatical errors and wrong spelling in the text that need extensive corrections.
